# A Dual-Process Model Applied to Two Health-Promoting Nutrition Behaviours

**DOI:** 10.3390/bs11120170

**Published:** 2021-12-08

**Authors:** Daniel J. Brown, Jessica Charlesworth, Martin S. Hagger, Kyra Hamilton

**Affiliations:** 1School of Applied Psychology, Griffith University, Brisbane 4122, Australia; danielbrown.psych@gmail.com (D.J.B.); jessica-paige.charlesworth@alumni.griffithuni.edu.au (J.C.); 2Menzies Health Institute Queensland, Griffith University, Gold Coast 4222, Australia; 3Psychological Sciences, University of California, Merced, CA 95343, USA; mhagger@ucmerced.edu; 4Health Sciences Research Institute, University of California, Merced, CA 95343, USA; 5Faculty of Sport and Health Sciences, University of Jyväskylä, 40600 Jyväskylä, Finland

**Keywords:** intention, habit, counter-intentional habit, nutrition, students

## Abstract

We tested a dual process model incorporating constructs that reflect both performing the target behaviour (behaviour directed habit) and habits that run counter to the target behaviour (opposing behaviour habit) in accounting for variance in two health behaviours: eating the recommended serves of fruits and vegetables a day and restricting sugar-sweetened beverage consumption. A prospective correlational design with two waves of data collection separated by one week was adopted. Participants (*N* = 606) comprising middle school students (*n* = 266) and university students (*n* = 340) completed an initial survey comprising self-report measures of past behaviour, intention, and habit to perform the target behaviour and habits that run counter to the target behaviour. One week later, participants (*N* = 414) completed a self-reported measure of behaviour. Results revealed that behaviour directed habits predicted fruit and vegetable consumption in both samples, while opposing behaviour habits predicted restriction of sugar-sweetened beverages in the middle-school sample only, with a moderating effect also observed. Current findings indicate that habits specifying avoidance of the target behaviour did not predict future behaviour. However, the moderating effect observed provides preliminary evidence that strong habits to perform a behaviour may override habit to avoid the behaviour.

## 1. Introduction

Health-promoting nutrition behaviours, such as consuming adequate fruits and vegetables and limiting foods and drinks with added sugars, is significantly associated with reducing chronic diseases [1]. Typically, dietary patterns are established in childhood [2] and remain relatively stable into adulthood [3]. Given the role of dietary behaviours in the prevention of chronic conditions, exploring the social psychological and behavioural determinants of health-promoting nutrition behaviours may provide formative evidence to guide more effective behavioural change interventions. This study, therefore, explored a dual process model, testing the effects of constructs representing both reasoned-action processes from models of social cognition (i.e., intention), and non-conscious processes (i.e., behaviour directed and opposing behaviour habits) on two nutritional behaviours (i.e., eating the recommended serves of fruits and vegetables each day and restricting sugar-sweetened beverages) in two samples (i.e., middle-school children aged 11–14 years and university students aged 17–24 years).

Behavioural scientists and interventionists seeking to understand and explain health behaviour, and in this context nutrition behaviours, have tended to apply theories of social cognition, like the theory of planned behaviour [4], to identify relevant determinants of nutrition behaviour [5,6]. These theories assume that behavioural engagement is determined by an individual’s intention to engage in the behaviour. Intention is viewed as the most proximal determinant of behaviour and has been shown to predict behaviour in numerous health contexts, including dietary behaviours [7,8,9,10]. Reviews of such theories indicate their constituent constructs explain, on average, around 20% of the variability in dietary behaviours [11], and around 45% of the variance in specific dietary behaviours, such as fruit and vegetable consumption [12]. Despite the effective application of these theories, there remains a number of unresolved issues with respect to the determinants of nutrition behaviours. For example, theories of social cognition typically specify determinants that reflect deliberative processes, with the underlying assumption being that behavioural engagement is governed by reasoned, conscious processes. However, emerging research suggests that including constructs that represent non-conscious processes, such as habit and behavioural automaticity, as parallel predictors of behaviour increase the variance explained in behaviours [13,14,15,16].

### 1.1. The Role of Non-Consciousness, Automatic Processes in Determining Behaviour

Dual process theories aim to provide more comprehensive means to explain variance in behaviour by incorporating constructs that represent non-conscious, automatic processes alongside constructs representing deliberative, reasoned processes [17,18,19]. Deliberative processes represent individuals’ reasoned deliberative consideration of the merits or concerns of performing a particular course of action before making a decision, often referred to as a system-2 process [20,21]. The effects of constructs of theories of social cognition on behaviour are often considered to reflect system-2 processes. Non-conscious processes are expected to reflect well-learned patterns of evaluations, contextual-cues, and behaviours stored in memory as schema that are activated on presentation of behavioural cues or behaviour-related information, and lead to efficient behavioural enactment with little deliberation [22]. Such behaviours are often considered to be enacted non-consciously, beyond the individual’s awareness, and are often referred to as a system-1 process. Habit is defined as a highly automated action generated in response to well-learned situational cues or specific contexts [23,24], and reflects a specific form of non-conscious process in dual process models.

Research supports the role of habits within dual process models applied to predict behaviour, including dietary behaviours [25,26,27,28]. Self-reported habit, in the context of dual process models, has been hypothesised to act as both a proximal determinant of future behaviour [19] and as mediator of the effects of other important behavioural determinants [24]. For example, habit, along with intentions, has been shown to mediate the relationship between the home environment and soft-drink consumption of adolescents [29]. Habit has also been hypothesised to act as a moderator of the intention-behaviour relationship. It has been argued that when individuals have strong habits, they no longer need to rely on intentions to enact behaviours [30]. In a meta-analysis, support was found for the moderating role of habit applied to dietary and physical activity domains; the intention-behaviour relationship was smaller when individuals reported higher levels of habit [31]. However, other studies have not supported this moderation effect [32,33,34,35].

There are, however, methodological arguments regarding the validity of whether habits lead to action when the intention-behaviour relationship is weaker. Gardner, Corbridge, and McGowan (2015) argue that, given most studies measure habit and intentions concurrently (e.g., habit to eat vegetables, intention to eat vegetables), habit would be expected to correlate with intention. They suggest that data which show strong habits and weak intentions or vice versa likely lack ecological validity [36]. Furthermore, although individuals may report weak intentions toward performing a given behaviour, they may hold strong intentions to perform an opposing behaviour [36,37], suggesting that interactions between habit and intentions should also be considered in light of habit toward performing the target behaviour as well as habits for the behaviour that runs counter to the target behaviour. For example, researchers testing the moderating effect of the habit to restrict sugar-sweetened beverages on the relationship between intention to restrict the beverages and behaviour should also consider moderating effects of habit to consume sugar-sweetened beverages. Measuring habits for behaviours that run counter to the target behaviour may be easier for participants to reflect upon in self-report measures. For example, participants may find it difficult to reflect on avoidance-oriented habits (e.g., the habit to avoid drinking sugary beverages) [38] or have not made specific cue-response associations for such an avoidance-habit. In these circumstances there may be a methodological advantage referring to habits that run counter to the target behaviour (i.e., habit to drink sugary beverages) [39,40].

Initial research investigating the role of habits that run counter to the target behaviour has revealed inconclusive results. Research into unhealthy snack intake among adolescents found that, while intentions for healthy eating were significantly associated with reducing unhealthy snack consumption, habit strength for eating unhealthy snacks was more strongly and positively associated with unhealthy snack intake [41]. So, this suggests that habits that run counter to the target behaviour may be more effective in explaining behaviour than intentions to be healthy. In contrast, another study found that habits to eat unhealthy snacks did not override intentions to avoid eating unhealthy snacks [36]. The latter finding suggests that exploring habits that run counter to the target behaviour can be inhibited in favour of intentions to perform the target behaviour. This may be dependent on available resources to overcome the opposing behaviour habit, such as knowledge and ability to use self-regulation skills to overcome the habit, which may maximize the strength of individuals’ intentions to perform the target behaviour [42,43]. However, inconsistent findings highlight the need for further research examining the circumstances in which habits to perform the target behaviour, and habits that run counter to the target behaviour, determine nutrition-related behaviours.

A further consideration is an individual’s past actions, as they are often found to be a consistent predictor of future behaviour. Past behaviour may reflect both intentional and automatic pathways to future behaviour [24]. However, including past behaviour in theories of health behaviour has been found to attenuate model effects [35,44,45,46,47], including for nutrition behaviours [7]. Such effects may reflect previous decision making and habits, as well as unmeasured constructs that may determine behaviour. The inclusion of past behaviour is, therefore, important to include in a dual process model. If the inclusion of past behaviour reduces the effects of the constructs reflecting either deliberative or non-conscious processes, possibly to the point of being comparatively trivial, then the theory should be considered insufficient as an as effective means to predict behaviour.

### 1.2. The Current Study and Hypotheses

The aim of the current study is to test a dual process model incorporating constructs reflecting deliberative processes (i.e., intention) and non-conscious processes (i.e., habit to perform the target behaviour (behaviour directed habit) and habits that run counter to the target behaviour (opposing behaviour habit)) for two distinct nutrition behaviours: eating the recommended serves of fruits and vegetables a day and restricting sugar-sweetened beverage consumption. The model is tested in two independent samples: middle school students aged 11 to 14 years and university students aged 17 to 24 years. The proposed model is presented in Figure 1. It is first hypothesised that, in accordance with theories of social cognition, intention will predict behaviour (H1). Drawing from theories on habit, it is hypothesised that behaviour directed habit (H2) and opposing behaviour habit (H3) will have direct effects on each behaviour. Furthermore, it is hypothesised that behaviour directed habit (H4) and opposing behaviour habit (H5) will moderate the intention-behaviour relationship. It is also expected that opposing behaviour habit will moderate the effect of behaviour directed habit on behaviour (H6). In line with other dual process models, it is expected that past behaviour will indirectly predict future behaviour via constructs reflecting both deliberative (i.e., intention, H7), and automatic (i.e., behaviour directed habit, H8; and opposing behaviour habit, H9) pathways. It is expected that including past behaviour will attenuate effects of theory relationships (H10) consistent with previous research. Last, we seek to compare the differences in the size of the effects of relations between the constructs of the proposed models across the two samples.

## 2. Materials and Methods

### 2.1. Participants

Participants in the school sample (*N* = 266; 45.9% female; M age = 12.61 years, SD = 0.61 years, range = 11 to 14 years) were year 7 and 8 students recruited from two co-educational schools (one public school and one private school) in South East Queensland, Australia. The index of community and socio-educational advantage (ICSEA), an indicator of socioeconomic status based on national census, was sourced for each school. The index was above the national average for the private school, while the index for the public school was slightly below the national average [48]. Participants in the university sample (*N* = 340; 73.2% female; M age = 19.22 years, SD = 1.88 years, range = 17–24 years) were first year undergraduate psychology students from a major university in Queensland, Australia. Using the software G*Power, a minimum sample size of 110 at time-2 was necessary to achieve power of 0.95 with a small effect size (0.1) with four predictors. We aimed to collect at least 25% more than the 110 minimum sample at time-1 to account for attrition at time-2.

### 2.2. Design and Procedure

Ethical clearance was obtained from the University Human Research Ethics Committee as well as the relevant school authority for the school sample (Ref No: 2018/244). All subjects and their parents (where relevant) gave their informed consent for inclusion before their participated in the study. To participate in the study, students needed to be within the defined age-range of 11–14 or 17–24 years old. A prospective correlational survey design was adopted, with participants completing self-report measures of the target behaviour and psychological variables at an initial point in time, Time 1 (T1), and measures of the target behaviour at a follow-up point, Time 2 (T2), one week later.

Schools were recruited via an email which provided an invitation to participate and information about the research. Following approval from school Principals, study procedures and resources were developed and administered in partnership with the school teaching staff. Participants were required to obtain parent/guardian consent prior to completing the surveys by providing a consent form signed by their parent/guardian. Participants provided consent to participate in the study by completing a consent item on the first page of the T1 survey. No incentives were offered to the school students or schools for participation. Participants from the university sample were recruited from a first-year psychology participant pool and via emails sent to the broader university student population. University students recruited via the psychology participant pool were provided partial credit for survey completion; however, no other incentives were offered to students recruited outside the participant pool.

A combination of online and paper-based surveys was used in the school sample, while the surveys used for the university sample were exclusively online. School students completed the questionnaire in class time for both time points. University students completed the questionnaire at a time of their choosing for both time points. Data across time-points for both samples were matched using a unique code identifier provided by participants.

### 2.3. Measures

Constructs were measured using previously validated multi-item psychometric instruments adapted to refer to target behaviours in the current study. Participants provided their responses on seven-point scales. Brief details of the measures are provided below, and a full set of measures are available in Appendix A.

#### 2.3.1. Target Behaviours

The target behaviours for the current research were restricting sugar-sweetened beverage consumption and eating the recommended serves of fruits and vegetables per day. The measures for these target behaviours were derived from the Australian dietary guidelines [49]. The guidelines, as well as examples of sugar sweetened beverages and standard servings of fruits and vegetables, preceded the survey measures in the T1 and T2 surveys to enhance participants’ understanding of the target behaviours.

#### 2.3.2. Intention

The measure of intention assessed at T1 was developed using standardised procedures set out by Ajzen (1991) [4]. Intention items for both behaviours used the stem: “Do you agree that in the next week…” which preceded three items (e.g., “I intend to [restrict my sugary-drink consumption/eat the recommended serves of fruits and vegetables per day]”).

#### 2.3.3. Habit

Behaviour directed habit was measured at T1 and was defined as habits consistent with performing the target behaviour (i.e., the habit of eating the recommended serves of fruits and vegetable and the habit to restrict sugar-sweetened beverage consumption). The automaticity subscale (i.e., the self-report behavioural automaticity index; [50]) of the self-report habit index [51] was used to measure behaviour directed habit for each behaviour. The respective common stems, “Restricting my sugary-drink consumption is something…” and “Eating the recommended serves of fruits and vegetables per day is something…” were followed by four items (e.g., “I do automatically”). Habit for behaviours that run counter to the target behaviour (i.e., opposing behaviour habit) was measured at T1 and was defined as the habit to perform the opposite of the target behaviour (i.e., the habit to drink sugar-sweetened beverages and the habit to avoid eating the recommended serves of fruits and vegetables). The same behavioural automaticity scale used to measure behaviour directed habits was used with changes in the common stem to reflect the opposing behaviour habit (i.e., “Drinking sugary-drinks is something…” and “Avoiding eating the recommended serves of fruits and vegetables per day is something…”).

#### 2.3.4. Behaviour

Standardised procedures set out by Ajzen (1991) [4] were used to develop a scale to measure behaviour for restricting sugar-sweetened beverage consumption and behaviour for eating the recommended serves of fruits and vegetables per day. For both behaviours the common stem, “Think about the past 7 days…” preceded two questions (“…in general, how often did you [restrict your sugary-drink consumption/eat the recommended number of serves per day of fruits and vegetables]?” and “…on how many days did [restrict your sugary-drink consumption/eat the recommended number of serves per day of fruits and vegetables]?”). Participants indicated their responses using seven-point scale (1 never/0–1 days and 7 always/7 days).

### 2.4. Data-Analysis

Variance-based structural equation modelling (VB-SEM) was used to test our hypothesized model. VB-SEM uses a partial least squares estimation method using a ‘distribution free’ estimation method. The analysis is less affected by model complexity or departures from normality than covariance-based methods [52]. Models were estimated using the Warp PLS v6.0 software [53]. Items from each instrument were used as indicators of latent variables representing each model construct in a structural equation model. Missing data were imputed using hierarchical regression imputation. All proposed paths among constructs detailed in Figure 1 were specified as free parameters in the model. In addition, we statistically controlled for the effects of age, ethnicity, gender, and past behaviour by setting these variables as predictors of all other variables in the model.

The measurement aspects of the model are used to assess the validity of the proposed measures. The loading of each indicator on its respective latent factor was expected to exceed 0.700. Composite reliability coefficients (ρ) and average variance extracted (AVE) statistics, which test the sufficiency of scale items as indicators the latent variables and whether the items account for sufficient variance in the factor, respectively, were expected to exceed 0.700 and 0.500. Discriminant validity was supported if the square-root of the AVE for each latent variable exceeded its correlation coefficient with other latent variables. Overall model fit was evaluated using multiple criteria: the goodness-of-fit (GoF) index with values of 0.100, 0.250, and 0.360 corresponding to small, medium, and large effect sizes, respectively; the average path coefficient (APC) and the average R2 (ARS), which should be significantly different from zero for an adequately-fitting model; and the average variance inflation factor for model parameters (AVIF) statistic, which should be less than 5.000 for a well-fitting model [53].

Multi-group analysis was used to test pairs of path coefficients in the hypothesised model across the two samples. The analysis calculates a ratio using the differences in path coefficients between two samples and the pooled standard errors for the specified path coefficients, as outlined in Kock (2018). The ratio produces a t-value and *p*-value for the comparison of each hypothesized path across the samples. Due to multiple comparisons, the critical value for p for the difference tests was set at 0.01.

## 3. Results

### 3.1. Participants and Attrition Analysis

Demographic characteristics of participants in each sample are presented in Appendix B. Seventy-five (28.2%) and 117 participants (34.3%) were lost to attrition across the two data collection occasions in school and university student samples, respectively. Attrition analysis for the school student sample indicated no differences in age (F(1,264) = 0.105, *p* = 0.746, η2 =< 0.001), ethnicity (F(1,264) = 0.310, *p* = 0.578, η2 = 0.001), or gender (F(1,264) = 0.214, *p* = 0.644, η2 = 0.001) between participants that remained in the sample and those lost to attrition. In addition, there were no differences between participants remaining and those lost to attrition on the automaticity, intention, and behavioural variables (Wilks’ Lambda = 0.985, F(1,263) = 1.02, *p* = 0.397, ηp2 = 0.015). Attrition analysis for the university student sample indicated no differences in age (F(1,335) = 2.290, *p* = 0.131, η2 = 0.07), ethnicity (F(1,337) = 0.549, *p* = 0.578, η2 = 0.002), or gender (F(1,338) = 0.900, *p* = 0.343, η2 = 0.003) between participants that remained in the study and those lost to attrition. There were also no differences on automaticity, intention, and behavioural variables (Wilks’ Lambda = 0.972, F(1,332) = 2.33, *p* = 0.056, ηp2 = 0.028).

### 3.2. Preliminary Analyses

The goodness of fit statistics revealed an acceptable overall fit of the model with the data according to the multiple indices adopted, as shown in Table 3 and Table 4. Measurement components of the VB-SEM confirmed that the latent variables met or approached criteria for construct and discriminant validity. Factor loadings for the latent factors exceeded the 0.700 criterion and AVE scores exceeded the 0.500 criterion supporting construct validity. Composite (ρ) reliability coefficients, AVE, and intercorrelations for model variables are presented in Appendix C. Reliability coefficients exceeded the 0.700 criterion, and AVE values exceeded the recommended 0.500 criterion. Correlations among the latent variables also indicated constructs achieved discriminant validity.

### 3.3. Model Effects

Restricting Sugar-Sweetened Beverages. Standardized parameter estimates for the hypothesized relations among factors are presented in Table 1. Overall, the model accounted for 40.5% of the variance of behaviour in the school student sample and 35.0% of the variance in behaviour of the university student sample. Results revealed statistically significant effects of intention on behaviour for both samples. Habit for the target behaviour did not predict behaviour in either sample. However, habit running counter to the behaviour negatively predicted behaviour in the school sample, but not the university sample. There were no significant moderation effects of either habits on the intention-behaviour relationship in either sample. However, opposing behaviour habit moderated the behaviour directed habit-behaviour relationship in the school sample. A review of the simple slopes (see Appendix D) revealed that as opposing behaviour habit increased, the behaviour directed habit was less predictive of future behaviour. No moderation effects was found in the university sample. There were statistically significant indirect effects of past behaviour on future behaviour via intention in both samples, but not through either habit measures. Multi-group analyses revealed no differences in the hypothesised relationships across the samples. Past behaviour significantly predicted all variables in the model. When past behaviour was excluded from the model, effect sizes of all model effects were larger, corroborating the attenuating effect of past behaviour on model effects observed elsewhere. Multi-group analyses between the model with and without past behaviour found no statistically significant differences in effects.

### 3.4. Eating the Recommended Serves of Fruit and Vegetables

Standardized parameter estimates for the hypothesized relations among factors are presented in Table 2. Overall, the model accounted for 79.5% of the variance of behaviour in the school student sample and 75.4% of the variance in behaviour of the university student sample. Results revealed statistically significant effects of intention on behaviour for both samples. Behaviour directed habit significantly and directly predicted behaviour in the school sample, but not the university sample. Opposing behaviour habit did not predict behaviour in either sample. There were no significant moderation effects in either sample. Past behaviour significantly predicted future behaviour via both intention and behaviour directed habit in both samples. No indirect effects were found via opposing behaviour habit in both samples. Multi-group analyses revealed significantly higher effects for the past behaviour on future behaviour relationship in the university sample. Past behaviour significantly predicted all variables in the model. When past behaviour was excluded from the model, the size of the model parameter estimates was larger and the relationship between behaviour directed habit and behaviour in the university sample was statistically significant, likely a suppressor effect. Multi-group analyses found the size of the relationship between behaviour directed habit and behaviour in the university sample was larger in the model excluding past behaviour. No other differences across the samples were found.

## 4. Discussion

Based on dual process models of action, the current study tested an integrated model in which constructs representing deliberative (i.e., intention) and non-conscious processes (i.e., habits for the target behaviour (behaviour directed habit) and habits that run counter to the target behaviour (opposing behaviour habit)) in two distinct nutrition behaviours: eating the recommended serves of fruits and vegetables a day and restricting sugar-sweetened beverage consumption. The proposed model was tested in two samples: a sample of middle school students aged 11 to 14 years and a sample of university students aged 17 and 24 years.

Consistent with previous theory and research (e.g., [4]), we found a significant, direct effect of intention on behaviour for each behaviour and in each sample. This suggests that middle-school students’ and university students’ intention to adhere to healthy eating practices, that is, eating the recommended serves of fruits and vegetables and restricting intake of sugar sweetened beverages, predicts self-reported behaviour. This demonstrates that irrespective of previous behaviour or current habits, intention to healthy eating practices is an important determinant of these nutrition behaviour. Such findings highlight intention as a potential target for intervention, and behaviour change techniques aimed at promoting intention could focus on strengthening intention formation by promoting change in the determinants of intention such as providing information on the value and benefits of performing the behaviour, which targets attitudes; highlighting the potential risks of performing, or not performing, the behaviour, which targets risk perceptions; and providing experiences of mastery, which targets perceived control and self-efficacy. Given the intention-behaviour relationship is small-to-medium in size, interventionists could consider using techniques such as making implementation intentions to bolster the strength of the relationship [54].

The current study also found direct effects of the constructs representing non-conscious processes on behaviour. Specifically, we found significant effects of behaviour directed habit for eating the recommended serves of fruits and vegetables in the middle school student sample. Thus, building strong cue-response associations may help promote consistent fruit and vegetable intake for middle school students. Interventions could focus on increasing the habit or automaticity for performing specific actions which may help habit formation, For example, making a visual cue (e.g., poster or note) that is left on the kitchen bench that reminds the participant to eat one piece of fruit with breakfast [55], or build on promoting the automaticity of a specific behaviour such as eating vegetable sticks with lunch every day [13,56]. Inclusion of past behaviour in the model in the university sample attenuated the effect of habit toward the target behaviour on the behaviour of eating the recommended serves of fruits and vegetables. This suggests that frequency of past behaviour accounts for the unique variation in future behaviour, over and above the contribution of the habit of eating fruits and vegetables. This attenuation effect suggests that a proportion of the effect of past behaviour in behaviour can be attributed to habit, which provides further evidence that past behaviour models habits. However, the large residual effect of past behaviour on behaviour suggests that a substantive proportion of the variance in behaviour is not attributable to habit. The residual effect of past behaviour may reflect effects of other unmeasured constructs in the model, such as effects of implicit beliefs or self-control [15,57,58,59].

Contrary to our hypotheses, no direct effects were found for the behaviour directed habit of avoiding drinking sugar-sweetened beverages. To speculate, one explanation for this finding may be that individuals find it difficult to reflect on whether they automatically restrict or avoid behaviours, as some of the contextual cues relevant to the habit are more ambiguous and less salient in memory [60]. Participants may have had uncertainty in judging the extent to which the behaviour was experienced as ‘unthinking’ and ‘automatic’. This is corroborated by previous research. For example, an exploratory think aloud study on self-report measures of habit, found that the most frequently occurring problem was that participants expressed uncertainty in their responses, which could be related to whether participants believe they habitually do not engage in a behaviour or simply do not engage in that behaviour [38]. Other research has suggested that mental accessibility of past behaviour moderates the intention-behaviour relationship [61]. It could be that accessing memories of specific instances of avoiding sugary-sweetened beverages is more difficult than approach-oriented behaviours, such as when an individual has consumed sugar-sweetened beverages habitually. Taken together, one possible reason for the lack of effects revealed here, and the overall inconsistency in the effects of self-reported habits across the research literature, may be due to excess error variance attributable to participants’ uncertainty in responding.

Further, we did not find consistent effects of opposing behaviour habit. We found opposing behaviour habit, that is, the habit of drinking sugar-sweetened drinks, predicted behaviour in the middle-school sample only. Thus, for the middle-school students, focused efforts on identifying the cues which stimulate the response to drink sugary drinks may be needed to further reduce this behaviour in future [62]. Middle-school students may have strong environmental cues (e.g., walking past shops on the way home from school, getting home from school and watching television) which might trigger the response to consume a soft drink. Intervention efforts could, therefore, be aimed at removing the availability of soft drinks in the family home to break the established mental link between the cue (getting home and watching television) with the response (consuming a soft drink) [63,64] or providing an alternative route home from school to avoid the cue (convenience store that has soft drinks) with the response (purchasing a soft drink). For university students, this could suggest that, unlike the younger middle-school students, habits for drinking sugary-drinks may not have a pervasive influence on behaviour, and, instead, intentions are more relevant. To speculate, a possible reason is that self-regulatory capacity may increase through childhood [65], so university students may have greater ability to exert self-control to adhere to healthy nutritional goals than their younger counterparts. This explanation is consistent with research demonstrating that self-control plays a more important role in regulating unhealthy but not healthy nutrition behaviours [66].

Opposing behaviour habit was also found to moderate the behaviour directed habit-behaviour relationship for middle-school students’ restriction of sugar-sweetened beverages. This suggests that as middle-school students’ habit to consume sugary drinks strengthened, so the effect of their habit of restricting their sugar-sweetened beverage consumption on restricting their actual consumption was attenuated. This demonstrates that, for this sample and behaviour, two distinct habitual processes exist and may be in competition with each other. However, it is important to note that this moderation effect was not found in any of the other samples or behaviours, and so, overall, current results do not provide strong evidence for this pattern of moderation. This effect may also depend on the level of specificity of the habit (e.g., the habit to eat cake when someone has offered it to you). Such a habit also has limited responses (e.g., to eat it or not to eat it). The habit strength of one response will likely be negatively correlated to habit strength of the other option. With more broadly specified habits (e.g., to exercise frequently) there are many potential cues and responses that will not necessarily always negatively be correlated with each other. Future replications studies may provide a better indication of the consistency of the effect.

Furthermore, habits should, theoretically, override intentions in guiding behaviour in associated settings [67]. Much of the research testing this hypothesis has used concordant intentions and habits, finding significant effects [31]. Given others have found null or opposing findings, it has been argued that the moderation effect only makes sense when using counter-intentional habits. The current investigation was the first to examine the simultaneous moderation effects of behaviour directed and opposing behaviour habits. Results found that neither of the two forms of habit moderated the intention on behaviour relationship across both samples and behaviours. Given the current study used similar methods and measures of habits, intentions, and behaviour to other studies, it calls into questions the validity of the expectation that habit will consistently override intentions in generating behaviour. It may be that habits will only override intentions under specific circumstances (e.g., an individual is preoccupied or is experiencing depleted self-regulatory capacity compared to feeling highly motivated to be healthy), or in certain contexts (e.g., eating in a social environment with friends compared to at home, alone). This result may also suggest that individuals engage in strategies that can inhibit habitual action. Studies have shown that students are aware of situations where habitually unhealthy behaviours are more likely to occur and can then choose to engage in strategies, such as monitoring and distraction, to inhibit the habit impulse [68].

Dual process models suggest that the stability of behaviour (i.e., past behaviours effect on future behaviour) is explained by both deliberative and non-conscious pathways [24,69]. Including past behaviour as a predictor in dual process models also acts as a test of sufficiency of the other determinants of behaviour in the model. The inclusion of past behaviour attenuated all model effects and reduced the effect of goal-direct habit on the restriction of sugar-sweetened beverages in university students to be not distinguishable from zero. Furthermore, the current research found mixed results for the role of both intentional and automatic factors in the mediation of past to future behaviour. In both samples and behaviours, past behaviour indirectly predicted future behaviour via intention. This demonstrates for middle school and university students, engagement in these nutrition behaviours appear to be in line with explicit goals and intentions. Unlike other research that found sun safety, oral-hygiene, and alcohol-consumption behaviours were each explained by factors representing automatic processes [14], the current research only found that past behaviour indirectly predicted future behaviour via behaviour directed habits for eating the recommended serves of fruits and vegetables in middle school students. Interestingly, while there was a significant, negative, direct effect of habits that run contrary to the behaviour on the restriction of sugar-sweetened beverages, this did not translate to a mediation effect. Therefore, while the habit of drinking sugary drinks impacts the likelihood of restricting total sugary-drink consumption, it does not explain the stability of the restriction behaviours.

A large amount of variance (approximately 75–80%) was explained in fruit and vegetable consumption compared to the restriction of sugar-sweetened beverages. One explanation of this could be the likely stability of contexts in which fruit and vegetable consumption occurs compared to the restriction of sugar sweetened beverages. Typically, the consumption of fruits and vegetables will occur regularly and in a stable situation (e.g., daily in the kitchen or dining room), whereas there may be greater variability in the contexts where participants restrict sugary drinks. Furthermore, there may be additional, unmeasured constructs that determine avoidance or inhibitory action, such as self-regulatory capacity and self-control. The model hypothesised in this research, therefore, accounts for a greater amount of variance in the behaviour that has smaller variability. This claim is substantiated by the strong effects of past behaviour on fruit and vegetable consumption compared to the restriction of sugar-sweetened beverages, demonstrating that fruit and vegetable consumption is relatively stable.

It is important to note, that for both nutrition behaviours in this study, effects of the habit measures framed as avoidance of the behaviour (i.e., the habit to avoid eating the recommended serves of fruits and vegetables or the habit to avoid drinking sugary drinks) were not statistically significant. This could suggest that either participants do not habitually avoid specific nutrition behaviours or participants struggle to consistently and accurately reflect on their habits to restrict or avoid certain behaviours. Furthermore, it could be that this construct is only relevant to certain people and the effect is lost within the “noise” of the data. For example, two individuals may never drink sugary beverages. Person A would previously consume sugary drinks frequently but several months ago formed the habit to avoid buying them. Person A therefore now habitually avoids/restricts their sugar-sweetened beverage consumption. A second person, Person B, has never consumed sugar-sweetened beverages and therefore has never tried to restrict their consumption; they simply do not drink them. Does this second person habitually not drink sugar-sweetened beverages or simply does not drink sugar-sweetened beverages? How should Person B answer the measure of habit? As habits are built over time, via repetition [39,56], conceptually, only Person A could have a habit to avoid sugary drinks but, as lay representations of habit and how lay-people answer measures of habit are not necessarily accurate [38] we do not know if individuals necessarily answer the question in line with the expectations of researchers. Furthermore, younger participants have less opportunities to develop avoidance habits and their food and beverage consumption is more likely to be under the control of external forces, such as their parents’ control. The two individuals might have different scores on the automaticity scale but have the same behavioural frequency, therefore losing the significant effect of behavioural automaticity on habit. This highlights the need to apply measures of habit that are relevant to the sample and behaviour, particularly with regard to avoidance-oriented behaviours, that are both conceptually meaningful and allow for consistent answering by participants.

### Strengths, Limitations, and Future Directions

The current study has several strengths including testing two distinct nutrition behaviours in two independent samples. The study tested two health promoting nutrition behaviours (i.e., fruit and vegetable consumption and the restriction of sugar sweetened beverages) in a middle school sample (students aged 11–14 years) and a young adult university sample (students aged 17–24 years). By using two behaviours and two distinct samples the consistency of effects can be evaluated, providing stronger evidence in support of the findings. Similarly, discrepancies between effects may provide insight into group level or behaviour-specific differences that could be further explored. This is the first study, to the authors’ knowledge, to simultaneously measure both behaviour directed habits and opposing behaviour habits. By including two separate automatic variables, this study was able to explore a nuanced representation of the roles that habit may play in nutrition behaviours.

Findings of the current study should be interpreted in light of a number of limitations, which are outlined next alongside suggestions for future research. There is a lack of research with regard to the concepts of habitual avoidance, restriction, or the habit of ‘not doing’ [70]. While the authors of this study attempted to use language and provide sufficient level of specificity when describing each measure of habit, there still remains concerns about how participants may have interpreted and answered these questions. There have been many concerns raised in the literature [71,72,73] regarding the measurement of habit in general; yet there remains a dearth of understanding about what the general population understand to be habits. Future research could focus on a deeper exploration of lay representations of habit to provide researchers knowledge about if and how measures of habit could be updated or refined to ensure they are measuring what is sought to be measured. Another limitation of this study includes the use of a prospective correlational design. Given the exploratory nature of the research this design is appropriate; however, it reduces the interpretability of the results as the direction of effects can only be inferred from the theoretically driven relationship structure. Future research should seek to use cross-lagged panel designs and experimental designs that manipulate the variables, to provide better evidence to support directions of effect and causality [74]. Last, the current study relied exclusively on self-report measures. This may introduce bias to results as socially desirable responding or inaccurate memory retrieval may inflate effects [75]. Where possible, future research should corroborate finding using non-self-report measures.

## 5. Conclusions

The current study tested a dual process model including constructs representing both intentional and automatic processes in two distinct nutrition behaviours (i.e., eating the recommended serves of fruits and vegetables and restricting sugar-sweetened beverage consumption) across middle school and university students. Results indicated that intention, representing a deliberative construct, significantly predicted both nutrition behaviours across both samples. There were inconclusive findings on the role of behaviour directed habit and opposing behaviour habit. Only the behaviour directed habit to eat the recommended serves of fruits and vegetables and the opposing behaviour habit to restrict sugar-sweetened beverages in the middle-school sample (i.e., the habit to drink sugary drinks), significantly predicted behaviour. The current study begins to explore the role of different forms of habit on health promoting nutrition behaviours. Furthermore, it highlights the need to further explore the concept of “opposing behaviour” habits. Future research should, therefore, focus on these areas including exploring lay representations, understandings, and use of “habit”. This will contribute to developing conceptually meaningful and accurate measurements of habit and to better understand the role both deliberative and automatic factors play in nutrition behaviour.

## Figures and Tables

**Figure 1 behavsci-11-00170-f001:**
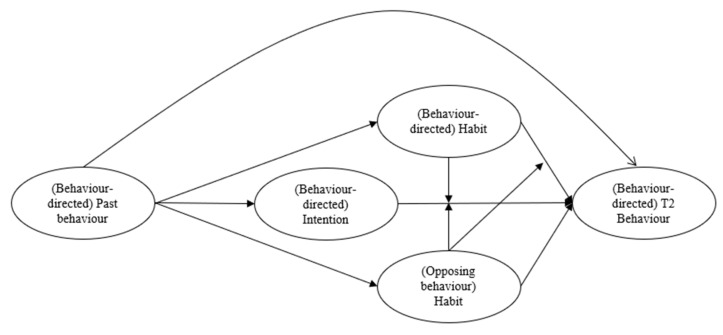
Hypothesized relations among model constructs. Direct effects of age, gender, and ethnicity on each model construct have been omitted for clarity. Standardized path coefficients for each effect are presented in Table 1 and Table 2.

**Table 1 behavsci-11-00170-t001:** Standardized Path Coefficients (β) and 95% Confidence Intervals from Structural Equation Models for the Restriction of Sugar Sweetened Beverages Between School and University Sample.

Effect	Restriction of Sugar-Sweetened Beverages without Past Behaviour						Restriction of Sugar-Sweetened Beverages with Past Behaviour					
	School Sample			University Sample			School Sample			University Sample		
	β	CI95		β	CI95		β	CI95		β	CI95	
		LL	UL		LL	UL		LL	UL		LL	UL
Direct Effects												
Intention → Behaviour	0.389 ***	0.258	0.520	0.424 ***	0.302	0.546	0.321 ***	0.188	0.454	0.299 ***	0.176	0.422
BDH → Behaviour	−0.054	−0.195	0.087	0.145 *	0.018	0.272	0.112	−0.027	0.251	0.072	−0.057	0.201
OBH → Behaviour	−0.223 ***	−0.358	−0.088	−0.005	−0.136	0.126	−0.177 **	−0.314	−0.040	0.010	−0.121	0.141
BDH X Intention → Behaviour	0.044	−0.097	0.185	0.055	−0.074	0.184	0.012	−0.129	0.153	0.092	−0.037	0.221
OBH X Intention → Behaviour	0.007	−0.134	0.148	−0.005	−0.136	0.126	0.006	−0.135	0.147	−0.039	−0.168	0.090
OBH X BDH → Behaviour	0.131 *	−0.008	0.270	−0.032	−0.163	0.099	0.118 *	−0.021	0.257	0.049	−0.080	0.178
Past behaviour → Intention	–	–	–	–	–	–	0.464 ***	0.335	0.593	0.574 ***	0.456	0.692
Past behaviour → BDH	–	–	–	–	–	–	0.430 ***	0.301	0.559	0.500 ***	0.380	0.620
Past behaviour → OBH	–	–	–	–	–	–	−0.373 ***	−0.504	−0.242	−0.426 ***	−0.548	−0.304
Past behaviour → Behaviour	–	–	–	–	–	–	0.273 ***	0.138	0.408	0.283 ***	0.158	0.408
Indirect Effects												
Past behaviour → Intention → Behaviour	–	–	–	–	–	–	0.149 **	0.051	0.247	0.171 ***	0.081	0.261
Past behaviour → BDH → Behaviour	–	–	–	–	–	–	0.048	−0.052	0.148	0.036	−0.056	0.128
Past behaviour → OBH → Behaviour	–	–	–	–	–	–	0.066	−0.034	0.166	−0.004	−0.096	0.088

*Note.* * *p* < 0.05 ** *p* < 0.01 *** *p* < 0.001; BDH = Behaviour directed habit; OBH = Opposing behaviour habit.

**Table 2 behavsci-11-00170-t002:** Standardized Path Coefficients (β) and 95% Confidence Intervals from Structural Equation Models for the Eating the Recommended Serves of Fruit and Vegetables Between School and University Sample.

Effect	Eating the Recommended Serves of Fruit and Vegetables without Past Behaviour						Eating the Recommended Serves of Fruit and Vegetables with Past Behaviour					
	School Sample			University Sample			School Sample			University Sample		
	β	CI95		β	CI95		β	CI95		β	CI95	
		LL	UL		LL	UL		LL	UL		LL	UL
Intention → Behaviour	0.558 ***	0.431	0.685	0.392 ***	0.062	0.270	0.460 ***	0.388	0.620	0.226 ***	0.101	0.351
BDH → Behaviour	0.289 ***	0.156	0.422	0.404 ***^1^	0.062	0.282	0.195 **	0.019	0.251	0.1021	−0.027	0.231
OBH → Behaviour	−0.044	−0.185	0.097	−0.081	0.066	−0.210	−0.095	−0.059	0.173	−0.036	−0.167	0.095
BDH X Intention → Behaviour	−0.003	−0.144	0.138	−0.088	−0.217	0.041	−0.033	−0.174	0.108	0.004	−0.127	0.135
OBH X Intention → Behaviour	0.065	−0.074	0.204	0.091	−0.038	0.220	0.109	−0.030	0.248	0.056	−0.073	0.185
OBH X BDH → Behaviour	0.051	−0.090	0.192	0.035	−0.096	0.166	0.045	−0.096	0.186	−0.050	−0.179	0.079
Past behaviour → Intention	–	–	–	–	–	–	0.671 ***	0.508	0.740	0.711 ***	0.595	0.827
Past behaviour → BDH	–	–	–	–	–	–	0.714 ***	0.605	0.837	0.755 ***	0.641	0.869
Past behaviour → OBH	–	–	–	–	–	–	−0.439 ***	−0.560	−0.328	−0.450 ***	−0.572	−0.328
Past behaviour → Behaviour	–	–	–	–	–	–	0.297 ***^a^	0.142	0.374	0.590 ***^a^	0.472	0.708
Past behaviour → Intention → Behaviour	–	–	–	–	–	–	0.308 ***	0.214	0.402	0.161 ***	0.071	0.251
Past behaviour → BDH → Behaviour	–	–	–	–	–	–	0.139 **	0.041	0.237	0.077	−0.015	0.169
Past behaviour → OBH → Behaviour	–	–	–	–	–	–	0.042	−0.058	0.142	0.016	−0.076	0.108

*Note.* * *p* < 0.05 ** *p* < 0.01 *** *p* < 0.001; ^a^ Significant difference (*p* < 0.05) between paths in the school and university sample; ^1^ Significant difference between paths with and without past behaviour; BDH = Behaviour directed habit; OBH = Opposing behaviour habit.

**Table 3 behavsci-11-00170-t003:** Model Fit and Quality Indices for Structural Equation Models for Restricting Sugar-Sweetened Beverages.

Behaviour	Restricting Sugar-Sweetened Beverages without Past Behaviour		Restricting Sugar-Sweetened Beverages with Past Behaviour	
Index	School	University	School	University
GoF	0.278	0.311	0.457	0.520
AR^2^	0.093 *	0.104 *	0.253 ***	0.294 ***
APC	0.107 *	0.090 *	0.142 *	0.133 *
AVIF	1.334	1.655	1.229	1.603

*Note.* * *p* < 0.05 ** *p* < 0.01 *** *p* < 0.001; GoF = Tenenhaus good of fit; AR^2^ = Average R-squared; APC = Average path coefficient; AVIF = Average full collinearity variation inflation factor.

**Table 4 behavsci-11-00170-t004:** Model Fit and Quality Indices for Structural Equation Models for Eating the Recommended Serve of Fruit and Vegetables.

Behaviour	Fruit and Vegetable Consumption without Past Behaviour		Fruit and Vegetable Consumption with Past Behaviour	
Index	School	University	School	University
GoF	0.434	0.423	0.684	0.695
AR^2^	0.207 ***	0.191 ***	0.514 ***	0.5156 **
APC	0.126 *	0.099 *	0.177 **	0.166 **
AVIF	2.182	2.227	1.496	2.531

*Note.* * *p* < 0.05 ** *p* < 0.01 *** *p* < 0.001; GoF = Tenenhaus good of fit; AR^2^ = Average R-squared; APC = Average path coefficient; AVIF = Average full collinearity variation inflation factor.

## Data Availability

Data and output can be found at https://osf.io/7yn53/?view_only=3a572a1637124bd49db932436efa6e6f.

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
