# Peer review of "A Dual-Process Model Applied to Two Health-Promoting Nutrition Behaviours"

_behavsci, 2021, doi:10.3390/bs11120170_

Round 1

Reviewer 1 Report

General comment: The research article entitled “A Dual-Process Model Applied to Two Health-Promoting Nutrition Behaviours” is a well-organized study, with sufficient methodology and adequate description of the results. Some minor corrections are required for the improvement of the manuscript.

Abstract: The Abstract is well written and adequately presents the aim and the basic results of the study.

-Authors should better define the study design and the type of the study. Is this a retrospective or a prospective study? Please define the duration of the follow-up If is prospective study.

Introduction: The introduction section adequately covers the need for investigation the health promoting nutritional behaviors.  

-Nevertheless, the introduction is too long and could be shorter in some cases, possible without sub-sessions.

-The 1.2 could be better incorporated into the study design at methodology session. The last paragraph of introduction should be a short description of the aim of the study.

Materials and Methods:  The materials and methods are adequately presented.

-Could authors describe the process for the selection of number of the participants? Did they use calculation program?

-Did authors use specific questionaries for medical history and nutritional habits for the participants? Please define.  

Results: The results of the study are analytically presented. Tables and figures are adequate explain the findings of the study.

-Preliminary data could not be included in order the results to be clearer and more specific.

Discussion: The results of study are sufficiently discussed.

Conclusion: The conclusion is adequate and summarizes the main text.

Bibliography/References: The references used by the authors cover adequately the relative scientific field and the aims of the study.

Author Response

We thank the reviews for their thoughtful comments regarding our manuscript. We have endeavoured to incorporate as many relevant changes as possible or provide further information to the reviews, as shown below.

COMMENT 1: Is this a retrospective or a prospective study? Please define the duration of the follow-up if is prospective study

AUTHOR RESPONSE 1: This is described in the abstract on lines 18 (“A prospective correlational design with two waves of data collection separated by one week was adopted”) and in section 2.2 from line 186 (“A prospective correlational survey design was adopted, with participants completing self-report measures of the target behaviour and psychological variables at an initial point in time, Time 1 (T1), and measures of the target behaviour at a follow-up point, Time 2 (T2), one week later.”)

These sections have been highlighted in text.

COMMENT 2: The 1.2 could be better incorporated into the study design at methodology session. The last paragraph of introduction should be a short description of the aim of the study.

AUTHOR RESPONSE 2: We thank the reviewer for this suggestion. While we agree that one way to present the aims and hypotheses of the manuscript is in the study design, our preference is to keep this information in the introduction too. We believe this is the natural conclusion to why we are doing the study, as outlined in the various gaps in the literature presented in the introduction.

COMMENT 3: Could authors describe the process for the selection of number of the participants? Did they use calculation program?

AUTHOR RESPONSE 3: We used G*Power to explore, a-priori, the required sample size. Using a small effect size (.1), an alpha of .05, power of .95, with 4 predictors, a total sample of 110 was necessary. We, therefore, wanted to ensure we had a minimum of 110 at time two for each of the two samples.

This information was added from line 180.

“Using the software G*Power, a minimum sample size of 110 at time-2 was necessary to achieve power of .95 with a small effect size (.1) with four predictors. We aimed to collect at least 25% more than the 110 minimum sample at time-1 to account for attrition at time-2.”

COMMENT 4: Did authors use specific questionaries for medical history and nutritional habits for the participants? Please define.

AUTHOR RESPONSE 4: No further medical information was included outside what is already described in the materials section.

COMMENT 5: Preliminary data could not be included in order the results to be clearer and more specific.

AUTHOR RESPONSE 5: Thank you for this comment. We believe that the preliminary analyses (section 3.2) is important so readers understand the validity of the model as per the various indices described.

Reviewer 2 Report

This manuscript seems to be in very good shape, and should be just about ready for publication. It addresses an important issue and does a good job on both substantive and methodological dimensions.

The authors should discuss the extent to which assumptions underlying the model approach have been satisfied.

Author Response

We thank the reviews for their thoughtful comments regarding our manuscript. We have endeavoured to incorporate as many relevant changes as possible or provide further information to the reviews, as shown below.

COMMENT 1: The authors should discuss the extent to which assumptions underlying the model approach have been satisfied

AUTHOR RESPONSE 1:

This data is included in the multiple indices presented in the preliminary analyses section (3.2).

“The goodness of fit statistics revealed an acceptable overall fit of the model with the data according to the multiple indices adopted, as shown in Table 1 and 2. Measurement components of the VB-SEM confirmed that the latent variables met or approached criteria for construct and discriminant validity. Factor loadings for the latent factors exceeded the .700 criterion and AVE scores exceeded the .500 criterion supporting construct validity. Composite (ρ) reliability coefficients, AVE, and intercorrelations for model variables are presented in Appendix C, supplemental material. Reliability coefficients exceeded the .700 criterion, and AVE values exceeded the recommended .500 criterion. Correlations among the latent variables also indicated constructs achieved discriminant validity.”

Reviewer 3 Report

The publication "A Dual-Process Model Applied to Two Health-Promoting Nutrition Behaviours" presented for review concerns an important issue that is in line with the current trends in nutrition.

However, it requires correction, especially in the editorial aspect, so that it can be published in Behavioral sciences:

- in affiliations, affiliation should first be 2, then 3 and 4,

- in affiliations, at e-mail addresses, there should be the authors' initials,

- references should be numbered in the order they are cited,

- the text should first refer to tables, and then they should appear in close proximity,

- line 155 - in the text there is a reference to figure 1, and in the publication it is 2 pages away,

- line 17 ".61 years"?

- headings 3 - should be in a different line than the text of the publication, the text should be justified,

- line 181 - the p level is set to "0.01" and hereinafter referred to as "0.05",

- line 294 - a space after "gender" is needed,

- tables - should be in accordance with the recommendations of the MDPI publishing house, e.g. table 1 may be arranged vertically, abbreviations should be expanded after the "-" sign, not "=",

- all abbreviations in tables should be expanded,

- line 327 and further - the text orientation should be vertical,

- title of table 5 - there is one unnecessary dot at the end,

- from line 362 - wrong page numbering,

- conclusions - should contain only the most important study summary,

- author contributions - should be in accordance with the specimen in the instructions for authors,

- funding - bad formatting,

- references should be after the "conflict of interest",

- references - must comply with the rules of the journal,

- why were these two populations selected for the study?

Author Response

We thank the reviews for their thoughtful comments regarding our manuscript. We have endeavoured to incorporate as many relevant changes as possible or provide further information to the reviews, as shown below.

COMMENT 1: Formatting issues:

- in affiliations, affiliation should first be 2, then 3 and 4,

- references should be numbered in the order they are cited,

- the text should first refer to tables, and then they should appear in close proximity,

- line 155 - in the text there is a reference to figure 1, and in the publication it is 2 pages away,

- headings 3 - should be in a different line than the text of the publication, the text should be justified,

- tables - should be in accordance with the recommendations of the MDPI publishing house, e.g. table 1 may be arranged vertically, abbreviations should be expanded after the "-" sign, not "=",

- all abbreviations in tables should be expanded,

- line 327 and further - the text orientation should be vertical,

- from line 362 - wrong page numbering,

- author contributions - should be in accordance with the specimen in the instructions for authors,

- funding - bad formatting,

- references should be after the "conflict of interest"

AUTHOR RESPONSE 1: We thank the reviewer for pointing out these formatting issues. We have addressed some of these (i.e., affiliation numbering and referencing), but refer to the journal editors for further formatting issues.

COMMENT 2: line 17 ".61 years"?

AUTHOR RESPONSE 2: We are unsure what the reviewer is asking of us here as this represented the standard deviation of age, as described in text.

COMMENT 3: line 181 - the p level is set to "0.01" and hereinafter referred to as "0.05",

AUTHOR RESPONSE 3: There are no p levels described on line 181. This may refer to the p level described in line 282. We adopted a more conservative p level for the multiple group comparisons to reduce the chance of a type 2 error.

COMMENT 4: line 294 - a space after "gender" is needed,

AUTHOR RESPONSE 4: This is now updated, thank you.

COMMENT 5: title of table 5 - there is one unnecessary dot at the end,

AUTHOR RESPONSE 5: This is now updated, thank you.

COMMENT 6: why were these two populations selected for the study?

AUTHOR RESPONSE 6: The specific populations were less important than having two distinct groups to be able to compare the consistency of model effects. We believe that having one child and one adult population would be a useful way to make these two groups distinct. Furthermore, we believed that there were likely important characteristics of these two groups that would make them important samples to study. For example, middle school students are likely to have their own preferences and intentions regarding fruit and vegetable and sugary drink intake, but also still be under the influence of others (e.g., parents).

Reviewer 4 Report

line 188 - what were the inclusion criteria for the study? were there any exclusion criteria? were the same criteria used in both groups?

table 3 is not needed in the text, the results presented will be discussed. It is too long and the presented results are not statistically significant, as the authors discussed in the introduction to Results.

Author Response

We thank the reviews for their thoughtful comments regarding our manuscript. We have endeavoured to incorporate as many relevant changes as possible or provide further information to the reviews, as shown below.

COMMENT 1: line 188 - what were the inclusion criteria for the study? were there any exclusion criteria? were the same criteria used in both groups?

AUTHOR RESPONSE 1: The only criteria we required was their age being in the specified limits. This has been added to line 184.

“To participate in the study, students needed to be within the defined age-range of 11 – 14 or 17 – 24 years”

COMMENT 2: table 3 is not needed in the text, the results presented will be discussed. It is too long and the presented results are not statistically significant, as the authors discussed in the introduction to Results

AUTHOR RESPONSE 2: We agree, and have now included this table as Appendix C.

Round 2

Reviewer 3 Report

The manuscript may be conditionally approved for publication. It is factually correct, but requires numerous editorial corrections.